# Diagnosis, Prevention, and Management of Fetal Growth Restriction (FGR)

**DOI:** 10.3390/jpm14070698

**Published:** 2024-06-28

**Authors:** Panagiotis Tsikouras, Panos Antsaklis, Konstantinos Nikolettos, Sonia Kotanidou, Nektaria Kritsotaki, Anastasia Bothou, Sotiris Andreou, Theopi Nalmpanti, Kyriaki Chalkia, Vlasis Spanakis, George Iatrakis, Nikolaos Nikolettos

**Affiliations:** 1Department of Obstetrics and Gynecology, Democritus University of Thrace, 68100 Alexandroupolis, Greece; k.nikolettos@yahoo.gr (K.N.); kotanidou.so@gmail.com (S.K.); nektaria.97@hotmail.com (N.K.); soterisand@hotmail.com (S.A.); theonalmpanti@hotmail.com (T.N.); kikichalkia@yahoo.gr (K.C.); spanakvls@outlook.com.gr (V.S.); nnikolet@med.duth.gr (N.N.); 2Department of Obstetrics and Gynecology Medical School, University Hospital Alexandra, National and Kapodistrian University of Athens, 12462 Athens, Greece; panosant@gmail.com; 3Department of Midwifery, School of Health Sciences, University of West Attica (UNIWA), 12243 Athens, Greece; natashabothou@windowslive.com (A.B.); giatrakis@uniwa.gr (G.I.); 4Department of Obstetrics and Gynecology, National and Kapodistrian University of Athens and Rea Maternity Hospital, 12462 Athens, Greece

**Keywords:** fetal growth retardation, prenatal diagnosis, prevention, management

## Abstract

Fetal growth restriction (FGR), or intrauterine growth restriction (IUGR), is still the second most common cause of perinatal mortality. The factors that contribute to fetal growth restriction can be categorized into three distinct groups: placental, fetal, and maternal. The prenatal application of various diagnostic methods can, in many cases, detect the deterioration of the fetal condition in time because the nature of the above disorder is thoroughly investigated by applying a combination of biophysical and biochemical methods, which determine the state of the embryo–placenta unit and assess the possible increased risk of perinatal failure outcome and potential for many later health problems. When considering the potential for therapeutic intervention, the key question is whether it can be utilized during pregnancy. Currently, there are no known treatment interventions that effectively enhance placental function and promote fetal weight development. Nevertheless, in cases with fetuses diagnosed with fetal growth restriction, immediate termination of pregnancy may have advantages not only in terms of minimizing perinatal mortality but primarily in terms of reducing long-term morbidity during childhood and maturity.

## 1. Introduction

Despite the significant reduction in perinatal morbidity and mortality over the last 30 years, fetal growth restriction (FGR), or intrauterine growth restriction (IUGR), is the second most usual cause of perinatal mortality [1]. Normal fetal growth is defined as fetal growth that corresponds between the 10th and 90th percentile of the growth chart, as defined specifically for different populations. The growth charts created for each population are represented with diagrams, where the normal growth of the crown–rump length (CRL) and the normal distribution of fetal weight are based on the gestational age for fetuses in 11–13^+6^ of gestational age [2]. Later in pregnancy, the growth charts also include the measurements of head circumference (HC), the biparietal diameter (BPD), the abdominal circumference (AC), the femur length (FL), and the ratio of head to abdominal circumference [2]. Fetal development depends mainly on genetic factors, with the fetal environment also playing a crucial role [3]. Also, it seems that the participation of insulin-like growth factors (IGF-I and IGF-II) and of a hormone involved in the metabolism known as leptin are also considered important in fetal development [4,5]. Finally, inadequate fetal growth is thought to be caused by disturbances in the metabolism of nitrogen oxides as a result of chronic acidosis [5]. It is important to accurately determine the gestational age by taking into consideration the last menstrual period and the crown-to-rump length of the fetus in the 11–13-week ultrasound scan. The evaluation of fetal growth relies on comparing the estimated fetal weight (EFW) with norms that are appropriate for the gestational age [6,7]. The development of the fetus is a dynamic and continuous process, and it takes a series of measurements with an interval of two-to-three weeks to determine whether it is normal or shows an abnormality. Fetal development is a highly complex collaboration between the hormones produced and the metabolism of the mother, the placenta, and the fetus. Important for fetal development is the sufficient vascularization and perfusion of the placenta. FGR is defined as the inability of the fetus to fulfill its genetic growth potential. The purpose of this literature review is to guide the methods of control, prevention, diagnosis, and especially the management of pregnancies with FGR, an important cause of perinatal morbidity and mortality.

## 2. Pathophysiology

Upon implantation of the zygote, the development of new vasculature and the proliferation of trophoblast cells commence. By the 10th week of gestation, a significant portion of the spiral arteries, which branch from the uterine arteries, undergo necessary adaptations to support pregnancy. Angiogenesis is an important factor in normal embryogenesis that involves the branching of new microvessels from pre-existing larger blood vessels, and it plays an important role in the development of the villous vasculature and the formation of terminal villi in the human placenta [4]. On the contrary, aberrant placental angiogenesis is associated with fetal growth restriction [4]. These vessels are small in diameter before pregnancy, with a hard muscular wall, and during pregnancy, they turn into larger-diameter vessels, with their diameter increasing up to 10 times, so that they can accept the greatly increased flow of blood and penetrate up to the inner 1/3 of the myometrium in order to cover the feeding needs. This remodeling also involves the replacement of the smooth muscle cells by syncytiotrophoblast cells [8]. In a percentage of women, these changes are not achieved, increasing the incidence of a complication during pregnancy [5]. The final part of these vessels does not expand as we would expect; the smooth muscle cells of their wall are not replaced by syncytiotrophoblast cells [8], as a result of which they remain rigid and cannot receive the increased amount of blood. Also, regarding penetration of these vessels into the myometrium, they do not manage to reach the desired depth. These findings are shown in pregnancies complicated by preeclampsia, FGR, recurrent miscarriages, and, in some cases, premature births [9]. In particular, the association between preeclampsia and altered angiogenesis is nowadays widely accepted [8,9].

### 2.1. Insufficient Trophoblastic Penetration

The exact reasons why the spiral vessels may not succeed in having the desired shape-conversion in pregnancy are not completely known. As for the main reasons to blame, it is, in principle, insufficient trophoblastic penetration. There are studies showing that substances released by the trophoblast during the penetration stage are necessary for the transformation of spiral vessels [10]. 

### 2.2. Immune Reactions

It also seems that the immune interaction between the pregnant woman and the fetus, with the entry of trophoblastic cells into the circulation of the pregnant woman, plays a role [11]. Studies have shown that a class of fetal trophoblast cells that have properties similar to natural killer (NK) cells are found at fairly high levels in the preterm and in cases with a specific type of histocompatibility (HLA-C) of pregnant women and may respectively facilitate or hinder the transformation of the spiral vessels. 

### 2.3. Biochemical Factors

Finally, various biochemical factors have been implicated, and mainly the angiogenic factors placental growth factor (PlGF) and VEGF, which seem to play an important role. Low levels of PlGF and high levels of PlGF-degrading receptor and VEGF occur after the end of spiral artery conversion [12,13,14]. In general, inflammatory reaction products, mainly cytokines, impair successful reproduction (prevent trophoblastic infiltration [15,16,17,18]. These changes in the supply of the placenta due to the relative ischemia due to the non-transformation of the spiral arteries in the expected state of pregnancy lead to a series of local and general changes [19,20,21,22]. The changes in the spiral vessels and the uterine arteries reduce the supply of nutrients mainly in the interlobular space and lead to the induction of oxidative stress, due to the increased oxygen radicals or the reduced amount of antioxidant substances occurring in cases of placental insufficiency. 

### 2.4. Apoptotic Pathways

In preeclampsia, placental changes related to the failure of the spiral arteries lead to the release of products into the maternal circulation that are responsible for the clinical picture [23,24,25,26]. In placentas from pregnancies that exhibit fetal growth restriction, the trophoblastic cells of the chorionic villi show an increased apoptosis in response to hypoxia and cytokines compared to placentas from pregnancies that have normal-for-gestational-age development [27,28,29,30,31,32]. Apoptosis is regulated at several levels. Trophoblast cultures exposed to hypoxia showed increased p53 activity, enhanced expression of the pro-apoptotic Mtd-1 [33,34], and decreased expression of the anti-apoptotic factor Bcl-2, all of which promote the process of apoptosis [31,32,33]. In contrast, only hypoxia, regulated by other proteins, such as the increased expression of the pro-apoptotic Bax and Bak, induces even more intense apoptosis [14,35]. Mtd-1, which increases in severe preeclampsia, is localized in syncytial nodules and is therefore thought to regulate their appearance in chorionic villi [33]. Whether Mtd-1 is elevated without the presence of preeclampsia has not yet been determined. In contrast, p53 expression is known to increase in the villi of FGR pregnancies compared to those of the control group, and this increase is evident in the histological examination of the chorionic villi [36,37]. Both histological and laboratory evidence support that p53 has a key role in regulating trophoblast apoptosis in pregnancies with residual fetal growth. Syncytial nodules (knots) are a hallmark of syncytiotrophoblastic apoptosis, and what distinguishes the generally increased apoptosis in FGR pregnancies compared to control group pregnancies is that these nodules in the first group are prominent [36]. In addition, the expression of caspase products in the sections reflects increased apoptosis in the villi of FGR pregnancies [28,34,36,37]. A characteristic of the villi of FGR is the cleavage of the intermediate filaments of cytokeratin 18 [38]. Cytokeratin 18 cleavage products and proteins with highly condensed nuclei act as markers of apoptosis on the surface of the villi and are particularly evident where there is a breakdown of syncytiotrophoblast continuity. Fibrin accumulations in the fibrin signal the resolution of syncytial continuity. This extensive damage to the villous trophoblast layer reduces functional syncytiotrophoblast mass in FGR pregnancies and limits the ability of the villi to contribute to nutrient transport. Furthermore, microscopic lesions functionally affect placental permeability, as α-photoprotein and small low-molecular-weight particles can pass into the feto-uteroplacental circulation without the mediation of syncytiotrophoblast cytoplasm and there is a correlation between elevated maternal serum alpha-fetoprotein and ischemic placental disease [39,40]. Either in response to or independent of this epithelial damage, an alteration in the normal balance of villous proliferation, differentiation, and apoptosis during their life cycle will limit the functional trophoblast mass of surface villi. However, the cytoplasm of the placentas of FGR pregnancies does not show a balancing increase in proliferation despite the apparently enhanced apoptosis of the trophoblast layer [30,32].

There is a wide variety of stimuli and mediators that possibly contribute to the damage seen in chorioallantoic villi, with oxidative stress being the main one [31]. One source of oxidative stress is placental ischemia from underdeveloped spiral arterioles. The production of free oxygen radicals (FORT) during oxidative stress is associated with tissue damage in many diseases [41], and trophoblast damage is caused more by hypoxia/reoxygenation than by hypoxia alone [42,43]. Placentas in FGR pregnancies show clear signs of oxidative stress, with decreased protein expression and a particular decrease in key proteins, including those in the AKT and mTOR signaling pathways [44]. The recognized dysfunction of trophoblast protein expression, signaling pathways, and life cycle provides important insight into some of the mechanisms by which oxidative stress causes the dysfunction seen in placentas of FGR [45]. Hypoxia or ischemia or both contribute to placental damage through mechanisms other than reactive oxygen species (ROS) production, as organ blood flow activates the complement cascade [14,18,31]. Dysregulated complement activation in non-pregnant women mediates immune damage in the heart, lungs, and kidneys. Evidence indicates that the complementary sequence also plays an important role in normal and non-normal pregnancies [39].

According to the findings of Baschat and colleagues, the membrane-attack complex c5b-9 (MAC) coexists in the concentrations of fibrous mass located at the sites of damage and is associated with apoptosis. Importantly, the number of fibrotic concentrations coexisting with MAC is remarkably higher in FGR pregnancies [40].

Abnormalities in the maternal spiral arterioles, impaired villous angiogenesis, and abundant concentrations of fibrin are characteristic in FGR pregnancies according to histopathological studies in the placentas of these pregnancies. When these changes are from hypoxia, ischemia/reoxygenation, complement activation, or another source, the chorionic villus layer reflects excess epithelial damage and endoplasmic reticulum stress (ER stress) indicative of disruption at the subcellular level.

The increasing gradient in placental dysfunction as it progresses during pregnancy results in limiting nutrient transport and decreasing blood flow to the fetus [40]. FGR is a condition where a fetus’s potential weight is not reached by the population owing to genetic or environmental variables, as assessed by genetics [11] and in association with updated birth weight, length and head circumference charts by gestational age. Often the terms “FGR” and “small for gestational age (SGA)” are used to describe the same problem, even though there is a distinct difference between them.

A neonate who is small for gestational age is a neonate whose birth weight, based on biometric assessment, is estimated to be less than 10% for the specific gestational age, gender, and several fetuses in relation to the normal weight for a specific population. FGR is a clinical definition for fetuses that fail to have the expected growth rate, with obvious signs of pathological reduction in their growth rate.

The main difference between SGA and FGR fetuses is that the small-for-gestational-age fetus may be small without an increased risk of adverse perinatal outcome, whereas the FGR fetus may have an estimated weight above the 10th percentile but an increased risk of perinatal morbidity and mortality [42]. The main differences between normal small fetuses and pathological small fetuses due to placental insufficiency or congenital abnormality are shown in Table 1.

## 3. Categories of Fetal Growth Restriction

There are three categories of FGR: symmetric, asymmetric, and mixed fetal growth restriction.

All measurements, including fetal weight, length, and head circumference, are reduced in symmetric FGR. The symmetric type of fetal delay begins early at the beginning of the second trimester. This type of fetal restriction usually is caused by endogenous factors, such as congenital infections or chromosomal abnormalities.

The asymmetric FGR begins at the end of the second or at the beginning of the third trimester. A reduction in the transport of nutrients with a limitation of glucose and fat storage, which comes mainly from the dysfunction of the placenta, presents in this type of restriction. 

Also, in asymmetric FGR, a reduction in the weight and size of the fetus due to brain sparing is presented. Subcutaneous abdominal fat, the aged appearance, and the loose skin folds manifest the reduction in nutrient transport [44,46].

In mixed FGR, a decrease in the number, as well as in the size, of cells is observed. Neonatal FGR brains show a significant reduction in mature IGL neurons and an 80% reduction in neuronal extension and branching, consistent with studies showing decreased cell mitosis and granule cell density in hypoxemic fetal sheep [45]. It is presented in advanced pregnancies when the fetal delay is additionally affected by placental causes. The clinical signs of the symmetric- and asymmetric-type presence of FGR are observed [44,46,47].

## 4. Causes of Fetal Growth Restriction

FGR is caused by maternal, fetal, genetic, or placental factors, with the majority of cases being due to placental dysfunction [48,49,50,51,52,53,54,55,56]. Multiple pathological maternal factors result in FGR. Usually, these conditions lead to a diminished uteroplacental perfusion, reduced oxygen-carrying capacity, or decreased nutritional transfer to the fetus [57]. 

### 4.1. Maternal Factors

#### 4.1.1. Maternal Age

Initially, both the young and the advanced maternal age increase the risk of FGR. A history of a previous complicated pregnancy by FGR increases the chance of recurrence by 25%, whereas mothers who have had small-for-gestational-age SGA babies have twice the risk of giving birth to FGR babies [57]. 

#### 4.1.2. Race and Socioeconomic Class

Furthermore, the race; the low socioeconomic class or living in a developing country, which contributes to poor nutrition and poor medical healthcare during pregnancy; and the alcohol consumption affect the normal fetal development [55,56,57]. Pregnant women living at high altitudes are exposed to chronic hypoxia, which affects birth weight. Studies in multiple countries have shown a direct association between high altitude and low birth weight [58,59,60,61,62]. FGR is more likely in cases when the mother’s birth weight, low weight before becoming pregnant, and low weight gain during pregnancy are all present [57,58]. It is still unknown how a hypocaloric diet or the insufficient consumption of specific nutrients may affect the development of FGR [59]. 

#### 4.1.3. Behavioral and Environmental Factor

The development of FGR fetuses is also influenced by a number of behavioral and environmental factors. Lastly, smoking, alcohol consumption, and drug abuse during pregnancy affect fetal development, according to research. Precisely, smoking during pregnancy leads to a 3.5% lower birth weight rate compared to not-smoking women, whereas up to 19% of low birth weight (LBW) babies are due to smoking during pregnancy. Another study showed that women who quit smoking 16 weeks before conception gave birth to newborns with a birth weight similar to that of the newborns of non-smokers [63]. Alcohol consumption affects fetal development and varies according to the amount and gestational age during consumption [64]. Fifty percent of heroin users and thirty percent of cocaine users during pregnancy deliver newborns with low birth weight [65,66]. Various medicinal substances, such as anticoagulants, folic acid antagonists, steroids, anticonvulsants, and antineoplastics, affect the fetal development, leading to FGR [67]. 

#### 4.1.4. Assisted Reproductive Techniques

Assisted reproductive techniques (ARTs) and infertility are considered independent risk factors for FGR. There is evidence that ovarian stimulation affects DNA methylation, causing changes in imprinting that can affect fetal development. Singleton neonates from IVF have an increased risk of LBW and SGA fetuses compared to non-IVF infants, whereas, in multiple pregnancies resulting from IVF, this does not apply [68]. 

#### 4.1.5. Medical History

Maternal pathological conditions also affect the pregnancy. For example, hypertension, preeclampsia, diabetes mellitus, systemic lupus erythematosus, chronic renal failure, and antiphospholipid syndrome affect microcirculation and reduce fetal perfusion, leading to hypoxia and subsequently to FGR [67]. Uteroplacental insufficiency associated with preeclampsia can be caused by the inadequate penetration of the trophoblast to the myometrium, resulting in persistent contraction of these vessels, subsequently atherosis, and blockage [69]. Diabetes mellitus, on the other hand, can cause hypoglycemic damage to the endothelial layer of both the micro- and macrocirculatory system, causing hypoperfusion and FGR [70]. Other medical conditions of the mother that affect the normal fetus development are autoimmune diseases and acquired thrombophilia [71]. In contrast, hereditary thrombophilia is not associated with FGR. Chronic maternal hypoxia as a consequence of pulmonary diseases (uncontrolled asthma and cystic fibrosis), cardiac diseases (cyanotic congenital heart disease and heart failure), or hematological disorders (severe anemia, β-thalassemia, hemoglobin H disease, and sickle cell disease) is associated with residual fetal growth [71,72,73]. Maternal gastrointestinal pathologies such as Crohn’s disease, spastic colitis, or gastrointestinal bypass may result in low-birth-weight neonates due to insufficient fetal nutrition. More precisely, protein deficiency or limited protein intake is associated with symmetric FGR. Additional maternal factors include factors from the uterus, such as myometrial fibroids [74,75] or periodontal disease [76]. A history of having an FGR infant in a previous pregnancy increases the chance of recurrent FGR by up to 25%. Mothers who have had SGA babies themselves have twice the risk of giving birth to FGR babies [77].

### 4.2. Factors Related to the Placenta

Placental insufficiency is a crucial factor in FGR and affects over 30% of all pregnancies; reduced placental circulation can affect the transport of oxygen and nutrients to the neonate, leading to FGR. The appearance of FGR is linked to the reduction in the mass and the function of the placenta [40,78,79,80,81,82,83,84,85]. Neonates diagnosed with FGR exhibit a placenta that is marginally lighter and approximately 24% smaller in volume when compared to placentas of healthy fetuses; in addition to the placental mass and function, immunological disorders of feto-uterine communication can result in FGR [78,79,80,81,82,83,84,85,86]. Activation of dendritic cells (DCs) at the site of embryo–placenta communication contributes to a better immune response of the fetus to support the development of the placenta and fetus [86]. These dendritic cells can be easily detected in peripheral blood, and their level of development is assessed in a normal pregnancy. A recent study by Cappelletti and colleagues points out that a reduced number of dendritic cells was detected in pregnancies with FGR compared to normal pregnancies. When confirmed, these dendritic cell changes at the embryo-placental surface may affect fetal and fetal vascular development during pregnancy [87]. Other common placental factors in FGR pregnancies are placental abruption, placental insufficiency, placenta previa, and placental hemangioma [80,88]. Limited placental mosaicism, single umbilical artery, and membranous attachment of the umbilical cord can also lead to FGR [67,88]. Rare placental tumors, such as chorioangioma, may cause a decrease in uteroplacental flow, with adverse effects on fetal development. Inflammation of the villi of unknown etiology has also been associated with FGR pregnancies [89].

### 4.3. Genetic Factors 

The decoding of maternal, placental, and fetal genes has shown that the polymorphism of these genes affects fetal development. Some of the genetic associations that are known so far are as follows. Homeobox genes: Homeobox genes were first discovered in Drosophila flies. These genes consist of 180 base pairs that encode a region of 60 amino acids, called the homeodomain. The homeobox genes that seem to be of particular importance in the placenta are DLX3, DLX4, MSX2, GAX, ESX1L, and HLX. There is reduced expression of HLX and ESX1L in cases of residual fetal growth [85]. SERPINA3 gene: The SNP located in the SERPINA3 promoter in the placenta also affects FGR [90,91]. NEAT1 (Nuclear Paraspeckle Assembly Transcript 1): The NEAT1 transcript is the main part of paraspeckles (the paraspeckles make up NAT1). This structure is responsible for keeping the mRNA in the nucleus. In a recent study by Gremlich et al., the expression of NEAT1 mRNA in residual fetal growth (FGR) was 4.14 times higher than in normal placenta, suggesting an increase in parastigma in damaging trophoblasts during residual fetal growth [92]. This could lead to increased retention of mRNA fragments in the nuclei of the damaged trophoblasts. Since damaged trophoblasts constitute an important functional barrier of the placenta, it could partially explain placental dysfunction in idiopathic FGR [92]. Placental growth factor (PlGF): PlGF plays an important role in human placental angiogenesis. Low concentrations of PlGF and high concentrations of its inhibitors (Fms-like tyrosine kinase-1) are associated with reduced angiogenesis [93]. 

Trophoblastic miRNAs (microRNA) in maternal plasma: microRNAs are small non-coding regions of RNA that regulate gene expression at the post-transcriptional level. miRNAs are released intracellularly into the circulation in pregnancy-complicated conditions. The concentration of specific miRNAs, such us miR-517c and miT-518b, is increased in the plasma of women with fetal development complications. These microRNAs play crucial roles in placental development and function. The dysregulation of these microRNAs can contribute to impaired placental function, leading to inadequate nutrient and oxygen supply to the fetus and, consequently, to FGR [94]. 

Apoptosis of Bcl-2 and Bax genes in human placenta: Bcl-2 belongs to the family of anti-apoptotic genes, while Bax belongs to the family of pro-apoptotic genes. Börzsönyi B and colleagues, in their study, evaluated Bcl-2 and Bax gene expression patterns in human placental samples from FGR pregnancies, using normal placentas. The study included 241 placenta samples (101 in the FGR pregnancy group and 140 in the normal pregnancy group). The Bcl-2 gene was expressed in both groups, while there was no difference between the two groups in the expression of the Bax gene. The degree of growth restriction within the FGR group was not associated with Bcl-2 and Bax gene expression. They concluded that the decreased inhibitory activity of the Bcl-2 gene and not the stimulatory activity of the Bax gene was responsible for the increased apoptosis observed in FGR [95]. Placental insulin-like growth factor (IGF1 and IGF 2) and the insulin-like growth factor binding protein (IGFBP)-3 genes: Börzsönyi B and colleagues, in the same study of 241 placenta samples, also evaluated the gene expression of the insulin-like growth factor-1 and the characteristics of glucose and insulin metabolism in the placentas of women with FGR. The research showed that the values of glucose and insulin in the umbilical cord of women with normal pregnancy were much higher compared to the values of women with FGR [95,96,97,98].

The same group of researchers compared the gene expression of epidermal growth factor (EGF) in 241 placenta samples. The findings suggest that variations in EGF expression may play a significant role in placental development and its associated complications [95].

Elevated endothelin-1 and leptin levels: Insufficient uteroplacental circulation is the most common reason for FGR. Endothelin-1 (ET-1), a vasoactive mediator, is mainly produced by endothelial cells and is a potent vasoconstrictor; together with its receptors, it is expressed in the human placenta. Ischemia is a potent stimulus for ET-1 production. The hormone leptin is a peptide that controls immunological responses, reproduction, and energy homeostasis, including both body weight and hunger. Adipose tissue is primarily where leptin is synthesized. Additionally, during pregnancy, the level of plasma leptin increases because leptin is produced by trophoblast cells of the human placenta. Nezar et al. investigated the role of leptin and endothelin-1 as indicators of prenatal growth restriction. Twenty-three of the forty-three women in their study had severe preeclampsia, and the remaining cases had non-preeclamptic FGR. Fifteen cases with normal pregnancies comprised the control group. The group of women with FGR [99] had a noteworthy rise in both maternal and fetal ET-1, according to the data.

Visfatin: Visfatin is a visceral adipose-specific lipocytokine and results in the expansion of fat storage in insulin resistance. Visfatin is identical to PBEF and has been found in both normal and infected human fetal membranes, and increased levels of it are observed during parturition. Malamitsi-Puchner et al. evaluated perinatal levels of visfatin (adipocytokine) in the growth-restricted endometrium. They measured serum visfatin levels in 40 mothers and their newborns 1-to-4 days after delivery. Twenty neonates had with FGR, while the rest had AGA. They found that visfatin levels were significantly higher in mothers who had FGR. Visfatin levels were also increased in FGR neonates [100].

Thrombophilia genes and FGR: Grandone and colleagues, in their retrospective study, observed a relationship between the birth weight of the neonates and the presence of factor V G1691A and factor II A 20210 mutations in mothers. They analyzed 755 women, including 194 with a history of unexplained recurrent pregnancy loss, 202 with gestational hypertension with or without proteinuria, and 359 with at least one normal pregnancy. Of 1100 neonates, they found 980 neonates from mothers without mutations and 136 neonates weighing less than 2500 g, and 34 of 123 mothers with a factor V G1691A or factor IIA mutation had a low-birth-weight infant [101].

Elevated levels of S100B protein: Florio et al. conducted a survey of protein levels in the urine of women with FGR. They evaluated 42 neonates with FGR and 84 neonates as a control group. They reported significantly increased levels of S100B protein in the urine of neonates with FGR and also reported that it had 95% sensitivity with 99.1% specificity in predicting adverse neurological outcomes in these neonates [96].

Genetic deletion of the growth factor Igf-1 and the SHOX gene: Caliebe et al. found deficiencies in the SHOX, IGF-1, and IGF1R genes in six neonates with FGR. They found gene variants in SHOX and the pseudoautosomal region (PAR). Another SHOX mutation is associated with a deficit in the IGF1R receptor, whose location was at Xp22.3 [97]. Genetic mutation in Igf1R: Kawashima et al. analyzed the nucleotide sequence of the IGF-1R receptor gene. They prepared cells of the receptor and identified mothers with FGR who had a heterozygous mutation in the L2 segment of IGF-IR. They hypothesized that this mutation leads to reduced cell division and reduces the effects of IGF [102,103].

## 5. Prenatal Diagnosis of Fetal Growth Restriction

It is really important to initially separate the fetuses with FGR from the SGA fetuses. The diagnosis requires the use of multiple diagnostic methods, such as weight growth curves; and Doppler parameters, including umbilical and uterine artery measurement, biophysical profile, and biometric indices [42].

Considerations for fetal size, including height, weight, maternal age and ethnicity, fetal sex, and number of pregnancies, are incorporated into normal weight distribution curves that are calculated for particular populations [104]. Adjustments to these variables have been suggested to identify small-for-gestational-age neonates [105].

FGR is predominantly attributed to placental insufficiency. The most common diagnostic method for the differential diagnosis of SGA and FGR is the Doppler of the umbilical artery, which evaluates the placental function. According to different Doppler patterns, FGR can be separated to asymmetric and symmetric types [106]. The role of Doppler in identifying the adaptation of the fetal cardiovascular system to hypoxemia is also important.

The failure of normal conversion of the uterine arteries from high- to low-resistance vessels is thought to reflect the deficiency of trophoblastic entry into the spiral arteries. This results in a high PI (PI > 95th percentile) in response to the vasodilation of the middle cerebral artery, which exhibits a reduced PI [107,108]. This effect is called brain sparing effect. Lastly, as an attempt to increase the blood flow to the heart of the fetus when severe oxygen deprivation is presented, absent or inverted a-wave is exhibited in the waveform of the DV [109]. Some individuals debate the fact that the absence or inversion of the a-wave of the DV is a consequence of increased intra-arterial pressure due to high cardiac load (increased placental circulatory resistance) and/or a direct consequence of fetal toxemia in myocardial cell function [110].

A combined evaluation of fetal muscle tone, respiratory movement, amniotic fluid volume, and fetal heart rate comprises the biophysical profile. An assessment of the biophysical profile can predict fetal pH, the presence or absence of fetal hypoxia, and ultimately the odds of perinatal death. The association between the change in the biophysical profile and the fetal pH seems to be stable at all gestational ages. Fetal pH levels below or equal to 7.20 are correlated with scores below 4. Fetal toxemia is characterized by a score below 2 [111].

The measurement of fetal heart rate via (cardiotocography) CTG helps to assess fetal oxygenation and, thus, placental function. Fetal hypoxemia is virtually ruled out by a reactive-type CTG recording of the heart rate. Fetal heart rate with short-term variation (STV) is a biophysical parameter of CTG that reflects the function of the autonomic nervous system. Fetal sympathetic or parasympathetic activity is altered in cases of FGR, leading to fetal rhythm differentiation.

Computed CTG and the assessment of fetal heart rate variability have been validated against the conventional CTG examination of fetal hypoxemia and toxemia and represent the only objective measure of fetal heart rate. An assessment of a compatible CTG does not provide the same information as a computed CTG, as the former is subjective because it relies heavily on interobserver and interobserver variability [112].

β-chorionic gonadotropin (β-HCG), pregnancy-associated plasma protein A (PAPP-A), placental growth factor (PIGF), and soluble tyrosine kinase-1 (sFlt-1) are some of the biomarkers that analysts are studying as potential biomarkers of placental dysfunction in early pregnancy. Their diagnostic value is very low when used individually. Nevertheless, research has demonstrated that, when combined with Doppler findings, they enhance the precision and sensitivity of cases of FGR pregnancies and preeclampsia [113,114].

The following recommendations are made by the International Society of Ultrasound in Obstetrics and Gynecology (ISUOG): Unless the estimated fetal weight (EFW) or abdominal circumference (AC) falls below the third percentile, the fetal size is insufficient to diagnose FGR. A decrease in growth velocity of more than 50% should alert the physician to FGR. Doppler measurement of the uteroplacental and fetoplacental circulation may be used to make a differential diagnosis of SGA and FGR. It is advisable to employ numerous diagnostic techniques when assessing pregnancies that are suspected to have FGR. Computed CTG and biophysical profiling should be used with Doppler measurements [42].

## 6. Monitoring Fetal Growth Restriction

FGR is one of the most common complications of obstetrics with increased chances of adverse outcome. Due to the absence of a cure for FGR pregnancies, the primary approach in managing these pregnancies is to evaluate the well-being of the fetus and determine the optimal time for delivery. Examinations for the well-being of the fetus can be distinguished into chronic and immediate tests. While the former will progressively have abnormal results due to hypoxemia and/or hypoxia, the latter are associated with abrupt changes that result in advanced stages and are characterized by severe hypoxia and metabolic acidosis and usually result in fetal death after a few days. Because there is no fixed sequence of fetal deterioration, it is necessary to recruit as many tests as possible to assess the condition of the fetus [115,116]. Guidelines from the Royal College of Obstetrics and Gynecology (RCOG) suggest that monitoring techniques and labor planning be incorporated into the management of FGR neonates [117].

The absence or reversal of end-diastolic flow is the most common finding of the symmetric pattern of FGR, and these patterns are reported to appear one week prior to acute fetal distress. Approximately forty percent of fetuses with acidosis exhibit this pattern on the umbilical artery Doppler [118]. The frequency of follow-up should be determined by the severity of fetal delay and abnormal umbilical artery Doppler.

Progressive worsening of the umbilical artery Doppler requires intensive monitoring every two or three days, especially when we have an absence or reversal of end-diastolic flow. The significance of this discovery remains unchanged, irrespective of the gestational week, in predicting perinatal morbidity and mortality [119,120,121,122,123,124].

The Doppler measurement of the middle cerebral artery is an early indicator of symmetric FGR cases. Up to 80% of fetuses present with vasodilatation two weeks before acute exacerbation [118]; however, in other studies, the percentage is exhibited in less than 50% of the cases [7,119,125,126,127]. Preliminary findings of acute middle cerebral artery vasodilatation in advanced stages of fetal risk have not been confirmed, and therefore, they do not appear to be relevant to the clinical management of FGR cases in early stages of pregnancy. On the other hand, in the identification of FGR in advanced pregnancy, there is evidence that vasodilatation of the middle cerebral artery is associated with an adverse outcome (fetal death) regardless of blood flow.

## 7. Prophylactic Administration of Corticosteroids and Magnesium Sulfate

The current recommendations for pregnancies with FGR advise the administration of corticosteroid prophylaxis as a preventive measure against respiratory distress syndrome in deliveries occurring before 34 weeks of gestation [128]. The Royal College of Obstetricians and Gynecologists (RCOG) recommends the administration of corticosteroids before the 36th gestational week [117].

Notably, despite these recommendations, there is no randomized trial to demonstrate whether the advantages of prophylactic corticosteroid use in preterm infants also apply to preterm infants with FGR. Regular monitoring is essential when administrating corticosteroids to fetuses with FGR which exhibit absent or reversed end-diastolic flow [129].

The efficacy of magnesium sulfate in neuroprophylaxis of preterm newborns has been demonstrated; however, the exact gestational age at which the impact diminishes remains uncertain. Although the administration of magnesium sulfate for neuroprophylaxis in FGR neonates is recommended by multiple studies, the timing of its initiation varies from before 29 weeks of gestation to even before 32-to-33 weeks of gestation [129].

## 8. Delivery Processing Time

As the only “treatment” we have to date for FGR pregnancies is delivery of the fetus, much thought is required regarding the appropriate timing of delivery, balancing the risk of iatrogenic morbidity and continued exposure of the fetus to an inhospitable endometrium environment. The TRUFFLE study demonstrated that using Doppler measures of the ductus venosus, electronic cardiotocography, and safety net criteria to determine the timing of delivery leads to better long-term neurodevelopmental outcomes for neonates who survive beyond two years. In order to achieve results similar to those of the TRUFFLE trial, it is necessary to use the same monitoring strategies and consider the same criteria for induction of labor. These criteria are based on the combination of ductus venosus Doppler and electrocardiogram findings [130].

According to the guidelines of the International Society of Ultrasound in Obstetrics and Gynecology, when a CTG scan is not available, the decision of elective delivery should be based on a combination of doppler (mainly ductal vena cava before 30 weeks’ gestation) and conventional CTG or biophysical profile. The presence of repeated automatic, unprovoked decelerations is an indication of labor. Nevertheless, when interpreting fetal activity on conventional CTG, gestational age and fetal maturity must be taken into account.

Likewise, an absolute indication for induction of labor is the mother’s condition (e.g., severe preeclampsia, eclampsia, and HELLP syndrome) or obstetric emergencies such as placental abruption. The close association between severe placental insufficiency and fetal hypoxemia and hypoxia is the absolute indication for planned caesarean section in most cases of early-onset fetal growth restriction [42].

On the other hand, in cases of late-onset FGR, there is no clear international guideline for the time of induction of labor [131]. In these pregnancies, when the umbilical artery pulsatility index is above the 95th percentile, experts recommend induction of labor beyond 36 weeks’ gestation and no later than 37 weeks plus 6 days [132]. Also, in FGR pregnancies with findings of cerebral redistribution of blood flow, the day of delivery is recommended to be planned around 38 weeks and no later than 38 weeks plus 6 days. Normal delivery is not contraindicated in cases of normal umbilical artery doppler, but continuous CTG is required [132].

## 9. Effects of Fetal Growth Restriction

The outcomes of FGR are determined by the degree of severity, the timing of beginning during pregnancy, and the cause of its development (genetic causes, maternal, placental, or neonatal causes). Furthermore, fetal growth restriction is associated with preterm birth and an increased risk of disabilities in surviving infants [133].

Inadequate prenatal nutrition and reduced-birth-weight neonates born with FGR are at a heightened risk for several pathological conditions, including hypothermia, glucose metabolic syndrome hypokalemia, polycythemia, jaundice, and sepsis. Additionally, there is a greater risk of infection, which may be correlated with a compromised immune system. Preterm infants born with FGR have a higher likelihood of developing retinopathy as a result of their early birth. FGR is associated with abnormal nephrogenesis due to hypoxia-induced abnormal tubular development, as well as abnormal lung development [87]. Factors involved in kidney loss include fetal hypoxia, reduced antioxidant content, and altered levels of growth factors. 

Conversely, clinical studies indicate that preterm FGR neonates have an elevated risk of developing bronchopulmonary dysplasia and respiratory distress syndrome in comparison to preterm infants with normal fetal growth, with long-term consequences such as bronchiolitis and asthma [134,135]. Furthermore, spirometry at school age may reveal diminished respiratory function in neonates with FGR or SGA. Studies have found a relationship between FGR and the onset of wheezing at 3 years of age regardless of gestational age at birth [136]. Additional research indicates that fetal growth restriction may have an effect on the immune system development of the neonate, due to the vulnerability to infections that they present [136,137]. 

FGR is known to induce notable alterations in the quantity of leukocytes and the immune response, in addition to impairing platelet count (thrombocytopenia) and red blood cell count (polycythemia). The number of B and T cells, number of neutrophils, and levels of G-antigens are lower in the umbilical cord of FGR neonates [137]. They also have a diminished number of T-regs compared to newborns with normal intrauterine growth and show some changes in size and the histopathology of the thymus gland [138].

The aforementioned alterations may contribute to postnatal complications, such as necrotizing enterocolitis, which frequently manifests in neonates born prematurely or with FGR [139]. Large epidemiologic studies highlight the association between fetal growth restriction and the risk of type 2 diabetes mellitus, obesity, hypertension, dyslipidemia, and insulin resistance leading to early cardiovascular disease [140].

The relationship between methylation, DNA acetylation, and histone differentiation permanently acquired variations resulting from fetal growth restriction, and its impact on gene expression in adulthood is supported by the Developmental Origins of Health and Disease (DOHaD) paradigm [141]. Potential complications associated with this hypothesis include circulatory and metabolic adaptations to accommodate the growing brain, such as short stature, early adrenarche, and the development of polycystic syndrome.

The risk of heart disease observed after FGR is associated not only with a higher incidence of metabolic syndrome but also with specific circulatory complications and dysfunctions. FGR is associated with high arterial hypertension and high heart rate and premature atherosclerosis. Abnormal thickening of the aortic and carotid walls detected in FGR fetuses and neonates may be the result of vascular remodeling before birth that may contribute to cardiac dysfunction in adulthood [142].

Furthermore, FGR neonates typically exhibit compromised renal anatomy and function, which elevates the risk of developing arterial hypertension and progressive renal failure [133]. FGR also affects brain development and function [143]. The fetal brain is particularly vulnerable to the effects of restricted fetal development and exhibits neurological disorders, such as cerebral palsy, epilepsy, learning disabilities, neurobehavioral disorders, and cognitive dysfunctions [144].

Infants born between 32 and 42 gestational weeks and weighing less than the 10th percentile for their gestational age are four-to-six times more likely to present cerebral palsy than infants born between the 25th and 75th percentiles [133]. In terms of spasticity and dyskinetic disorder, or ataxia, similar frequencies are noted in these infants. The relationship between birth weight and the chance of cerebral palsy is less evident in babies born before the thirty-second week of pregnancy.

Reports show that cerebral palsy increases up to 30 times when the fetal delay is of genetic etiology [142]. Approximately 75% of brain lesions associated with cerebral palsy occur in the early or mid-third trimester, a time of FGR and an adverse fetal environment, according to MRI studies [145]. In addition, studies that followed FGR neonates up to school age presented a significant link between fetal growth restriction and neurodevelopmental disorders, ranging from mild cognitive impairment to severe neuropsychiatric disorders [146]. A comparison between monozygotic twins with FGR and twins with normal development at school age showed that the former have an increased risk of cognitive impairment and low verbal IQ [147].

The timing of the onset of FGR and its severity influence its neurodevelopmental consequences [147].

Studies have shown the presence of severe white matter lesions that persist into adulthood, while mild fetal growth restriction is associated with only transient hypomyelination, mild microglial activation, and astrogliosis, with behavioral disturbances noted by the 8th week after birth [148].

Preterm birth usually exacerbates neurodevelopmental disorders resulting from fetal growth restriction [146]. Research has demonstrated that infants born between 29 and 32 weeks of gestation who have fetal growth restriction, even in moderate cases, have a significantly greater incidence of neurocognitive and behavioral disorders compared to children with a normal weight for their gestational age [149].

For this reason, low birth weight has recently been suggested as a factor that may help predict the occurrence of cerebral palsy in preterm infants [150]. During chronic hypoxia in the prenatal period, fetal blood flow is selectively redirected to the brain to increase oxygen and nutrient delivery. Although the brain sparing mechanism exhibited by these fetuses was initially thought to be protective, neurodevelopmental outcomes are better in fetuses with residual fetal growth that do not exhibit this mechanism [144].

Changes occurring in various parts of the brain, such as the frontal lobe or basal ganglia, to improve blood flow due to hypoxemia lead to neurobehavioral disorders [151]. Gray matter volume in the cerebral cortex is reduced by 28% in children with FGR compared to age-matched children born at term. Delayed development of the cerebral cortex and differentiated cortical cellularization are described in FGR neonates soon after birth and are found in cases of severe developmental disorders up to 1 year of age [152].

Disturbances in cell proliferation and loss of neurodevelopment are also seen in the hippocampus. These morphological changes observed in the hippocampus are likely to be responsible for hippocampus-related behavioral disorders related to learning and memory [147]. In addition, defective myelination and aberrant connectivity are common features in preterm infants with residual fetal growth [153,154].

Mechanisms such as excitotoxicity, oxidative stress, necrotic and apoptotic degeneration, and neuroinflammation are being investigated for involvement in brain damage associated with fetal growth restriction. Hypoxia and malnutrition activate a sequence of cellular and biochemical reactions that lead directly to cell death or lead to delayed cell death, with potential effects on immature neurons and ganglia [155].

Neuroinflammation is the key element in disturbed oligodendroglial maturation [156]. In cases of fetal growth restriction caused by prenatal undernutrition, a transcriptional analysis at birth shows dysregulation of genes controlling neuroinflammation in microganglia and oligodendrocytes [157]. These findings support the dual-damage hypothesis in the human brain: initially the damage is associated with the onset of FGR, which in turn is deleterious, and with postnatal effects such as systemic inflammation and epigenetic changes [158].

The development of neonates with FGR depends on the cause of their FGR, the coverage of their nutritional needs after birth, and also their social environment. Neonates with symmetric FGR remain small throughout their lives. On the other hand, newborns with the asymmetric type of growth delay will be able after their birth to reach the average levels of growth and achieve their inherited growth potential if they are provided with a favorable environment and sufficient nutrients and their caloric needs are met [159].

## 10. Prevention of Fetal Growth Restriction

The prevention of FGR requires interventions both before and during pregnancy. While some attempt to enhance the health of the mother, others are concerned with preventing placental insufficiency and ameliorating placental conditions [160,161]. The health of the mother and her diet are among the factors responsible for the occurrence of FGR. The concentrations of micronutrients, including vitamins, trace elements, and minerals, in the maternal diet can have an impact on the pregnancy outcome. Micronutrient supplementation is particularly beneficial in underdeveloped nations, where mothers frequently experience malnutrition. More precisely, a systematic review conducted by Haider presented that the administration of multivitamins has a better outcome in contrast to the administration of only one or two nutrients [162]. On the other hand, another study found no clinical effect on placenta or neonatal weight associated with the administration of multivitamins.

Only vitamin C appeared to have any positive association with placenta and birth weight [163]. However, multinutrient supplementation does not appear to prevent the onset of fetal growth restriction or the birth of small-for-gestational-age neonates [162].

Kramer et al., in a systematic review of experimental research on balanced protein intake in pregnant women, reported a reduction in SGA neonates, as well as in rates of FGR and sudden fetal deaths [68].

A Cochrane systematic review of data comparing routine or no antimalarial medication for malaria prevention in endemic areas concluded that routine administration in pregnant women reduced postpartum parasitemia-increased birth weight, low-for-gestational-age birth weight by 43%, and postnatal anemia with no effect on the number of perinatal deaths or the occurrence of FGR [164]. Aspirin has multiple effects at the vascular level, and therefore it is considered that it may contribute to the prevention of FGR. Most studies of aspirin look at its effect on the occurrence of preeclampsia first and secondarily on the occurrence of FGR. In 2018, two concurrent systematic reviews [165,166] confirmed existing evidence that aspirin provides a small reduction in the risk of FGR and SGA neonates depending on the age of initiation. Information from the first review looking at individual FGR case data supports starting aspirin before 16 weeks’ gestation where permissible [166]. The second review suggested a dose–response relationship in low-birth-weight infants. In these cases, the administration started before or during the 16th week of pregnancy, and the ideal dose was 100–150 mg [165]. Studies show circadian effects of aspirin on plasma renin activity and urinary excretion of cortisol, dopamine, and norepinephrine. Clinical trials also show the circadian effect of aspirin in the treatment of mild hypertension in non-pregnant women and recommend the prophylactic administration of aspirin to prevent preeclampsia. Two small randomized trials in pregnant women found that taking aspirin in the evening rather than in the morning was associated with a reduction in pressure and the occurrence of FGR [167].

In high-risk pregnancies, the majority of national and international guidelines prescribe a dose of 100–150 mg of aspirin to avoid FGR and SGA newborns. Pregnant women are frequently given unfractionated heparin and low-molecular-weight heparin for thromboprophylaxis and venous thromboembolism treatment. Both of these types of heparin do not cross the placental barrier and therefore do not directly endanger the fetus. Low-molecular-weight heparin is now used for thromboprophylaxis because of its improved efficacy and perceived safety [168]. Because of its anticoagulant qualities, heparin was initially used to prevent placental pathology and clots in the placenta that cause miscarriages [169,170].

## 11. Management 

In cases where a congenital abnormality of the fetus due to a genetic or infectious factor is found, the continuation or termination of the pregnancy is decided based on the gestational age and legislative restrictions of each country, the severity of the abnormality, the expected prognosis, and the wishes of the parents; the wishes of the parents are considered after full and clear information is given to them by a specialist doctor [171].

In cases of FGR due to placental factors, FGR is defined as early or late. Early FGR develops before 32 weeks, and the umbilical artery Doppler is the first clinical sign that appears abnormal, and the decision to terminate the pregnancy is balanced between the complications of prematurity and FGR with the risk of intrauterine death. Late FGR, which presents after the 32nd gestational week, does not show dramatic changes at the Doppler of the umbilical artery [172]. The pathological Doppler of the middle cerebral artery is an indication of persistent fetal hypoxia, wherein the artery adjusts blood flow in an effort to compensate for the embryo. Regardless, follow-up on pregnancies involving FGR must be conducted in a facility with the necessary resources and a neonatal intensive-care unit to manage high-risk pregnancies.

### 11.1. Monitoring the Fetal Well-Being

FGR pregnancy monitoring begins with training to recognize and count fetal movements by the mother. Fetal movements are diminished when they are less than 10 in a two-hour period. Although not a dependable method of FGR pregnancy monitoring, maternal fetal-movement counting remains a practical external method of pregnancy control [172].

The CTG method verifies the “well-being” of the fetus at the time of examination and has no prognostic value. Reassuring CTG includes normal baseline and variability and the presence of more than two accelerations. In contrast, non-reassuring CTG is characterized by the absence of accelerations in a 40-minute recording. The most important parameter in the assessment of CTG is variability. The frequency of application of the examination is recommended to be biweekly, with the exception of cases of low amniotic fluid or low biophysical profile, without determining the most appropriate schedule for repeating the examination. It is prudent, therefore, in these cases to individualize the frequency of the CTG according to the judgment of the attending physician or the protocols of each clinic [172,173,174,175].

Computed CTG, when available, allows for the estimation of short-term variability and its quantification in ms. Short-term variability is characterized as pathological when it is less than or equal to 3 ms, so it is an indication of fetal metabolic acidosis. CTG is evaluated alone or in conjunction with amniotic fluid assessment. The assessment of the biophysical or modified biophysical profile is the smaller application in Europe and the permanent method of pregnancy control in the USA and Canada.

A modified biophysical profile of less than 6/10 is considered seriously pathological and requires intensive monitoring until the achievement of optimal fetal maturation or completion of the 34th week of pregnancy. For a score of less than 4/10 and a pregnancy greater than 32 weeks, the pregnancy should be terminated, while for a pregnancy less than 32 weeks, intensive monitoring is not certain to offer the best results compared to termination of pregnancy.

### 11.2. Doppler 

The Doppler of the umbilical artery is the control indicator of the placental blood circulation. With an increased PI, absence or inversion of the end-diastolic flow wave of the umbilical artery are, in order of severity, and an indication of the disturbance of the uteroplacental circulation, there is an increase in the severity of the intrauterine delay of the fetus and a possibility of intrauterine death [172,173,174,175]. A recheck of the normal PI of the umbilical artery in pregnancies with SGA fetuses should be performed weekly or every two weeks. In the case of FGR with an increased PI or AEDV (absence of end-diastolic velocities), monitoring should be performed twice a week at least. For REDV (reverse of end-diastolic velocities), monitoring is recommended to be daily or at least three times a week [172,173,174,175].

The middle cerebral artery (MCA) Doppler is suggested as a subsequent technique for late FGR. In addition, synchronous Doppler monitoring of the umbilical artery allows for control of the cerebroplacental (CPR) quotient. Both MCA-PI and CPR appear decreased, reflecting placental insufficiency and subsequent fetal hypoxemia, even in cases where the PI of the umbilical artery (UA-PI) is unaffected. The CPR index is more strongly associated with the level of fetal hypoxia compared to the PI of the middle cerebral artery, as well as of the umbilical artery, when considered as separate measures [172,173,174,175].

The utility of the PI of the middle cerebral artery and CPR is important, especially in cases in which the PI of the umbilical artery appears normal. Pregnancy monitoring with DV Doppler should be performed in centers with a high-risk-pregnancy-monitoring department and well-structured NICUs to achieve the best possible perinatal outcome. An abnormality in the DV Doppler occurs when there is an increase in the PI. Its worsening occurs progressively within 48 h with absence and then, finally, reversal of the alpha wave. Abnormal values or waveforms predispose to the appearance of abnormal CTG and biophysical profile, as well as being a strong predictor of impending intrauterine death. The pulse of DV Doppler should be twice a week upon absence of end-diastolic velocities in the umbilical artery and three times a week when reverse of end-diastolic velocities in the umbilical artery is presented. Abnormal DV Doppler demonstration increases the need for closer monitoring if the requirements for terminating the pregnancy are not met. The criteria include the labor of a fetus presenting early fetal growth restriction or the absence/reversal of the umbilical artery end-diastolic flow wave. 

In the case of normal DV Doppler, labor will be planned after the 32nd gestational week. In the event of pathological DV, emergency labor should be initiated without delay [172,173,174,175].

### 11.3. Management Based on Gestational Age and Doppler Findings

This algorithm provides a comprehensive approach to managing FGR pregnancies, ensuring regular monitoring, timely interventions, and appropriate delivery planning to optimize outcomes for both the mother and the fetus [117,176]

<24 Weeks:
○Detailed fetal anatomy scan.○Consider referral to maternal–fetal medicine specialist.○Discuss prognosis with parents.
24–32 Weeks:
○Normal Doppler: Continue monitoring with ultrasounds every 2 weeks.○Abnormal Doppler:○Absent or reversed end-diastolic flow (AREDF) in umbilical artery: Monitor twice weekly with NST/BPP and Doppler studies.○Middle cerebral artery (MCA) Doppler: Consider delivery if MCA pulsatility index (PI) < 5th percentile.
32–37 Weeks:
○Normal Doppler: Continue monitoring with ultrasounds every 2 weeks.○Abnormal Doppler:○Absent or reversed end-diastolic flow: Consider delivery at 32–34 weeks.○Middle cerebral artery Doppler: Consider delivery if MCA PI < 5th percentile.
>37 Weeks:
○Normal or Abnormal Doppler: Consider delivery if fetal growth has plateaued or there is evidence of fetal compromise.
Interventions
○Maternal Interventions:○Recommend cessation of smoking and control of maternal diseases (e.g., hypertension and diabetes).○Consider aspirin for women at high risk of preeclampsia.
Fetal Interventions:
○Antenatal corticosteroids for lung maturity if delivery before 34 weeks is anticipated.○Magnesium sulfate for neuroprotection if delivery before 32 weeks is anticipated.
Delivery Planning
○Timing of Delivery: Individualize based on Doppler findings, fetal condition, and gestational age.○Mode of Delivery:
▪Vaginal delivery preferred if no contraindications.▪Consider cesarean delivery for severe FGR with abnormal Doppler findings or if fetal distress is present.



## 12. The Role of Cerebroplacental Ratio

Many studies indicate that a low CPR suggests a redistribution of fetal blood flow, commonly referred to as the brain-sparing effect, and is associated with negative outcomes for neonates. These adverse outcomes can include abnormal growth rates, NICU admissions, emergency cesarean sections due to fetal distress, intraventricular hemorrhage, hypoxic-ischemic encephalopathy, necrotizing enterocolitis, bronchopulmonary dysplasia, sepsis, and even mortality [177,178,179].

Infants born at term and weighing above the 10th percentile for gestational age are categorized as Appropriate for Gestational Age (AGA). While AGA fetuses are typically healthy, a subset of them may experience placental insufficiency and not achieve their optimal genetic growth potential [180,181,182]. AGA fetuses, especially those experiencing late-onset growth retardation after 34 weeks of gestation and exhibiting an abnormal CPR, are at increased risk of fetal distress during active labor, possibly necessitating an emergency cesarean section. In such instances, a decrease in cord blood pH and a higher rate of NICU admissions are observed compared to fetuses with a normal CPR [182].

CPR serves as an initial diagnostic tool for identifying pregnancy complications and can be evaluated alongside biophysical profile scores and UA/MCA Doppler studies. It stands as an independent metric for assessing third-trimester fetuses, irrespective of UA and MCA results [182].

The risk of stillbirth in prolonged pregnancies has been linked to late placental insufficiency and fetal hypoxemia. Fetal Doppler monitoring can enhance the management of fetal conditions, although findings in the literature vary. CPR is effective in detecting fetal hypoxemia through mechanisms such as reduced resistance in the MCA (brain-sparing effect) and increased resistance in the placenta. It is considered more critical than Doppler indices of the MCA and UA alone in predicting adverse fetal outcomes related to growth restriction and extended gestation [183,184,185].

The literature extensively discusses the advantages and drawbacks of calculating the CPR and its implications. Several studies have correlated CPR with growth rates and birth weights [177,186,187,188,189,190,191,192,193,194,195,196,197,198,199].

Furthermore, research has demonstrated that CPR is a significant independent predictor of stillbirth and perinatal mortality, with a high sensitivity for predicting abnormal fetal heart rates during labor and adverse neonatal outcomes in prolonged pregnancies [181,182,190]. However, some researchers argue that there is no significant correlation between CPR and pregnancy outcomes in extended pregnancies [182].

It has been suggested that fetuses with a CPR below 0.67 face an increased risk of intrapartum complications and are less likely to be delivered vaginally [191]. These findings align with our study’s results, which indicate that CPR is indicative of compromised fetal growth velocity and negative pregnancy outcomes, even in fetuses considered of appropriate size based on standard biometric assessments [192].

## 13. Future Therapeutic Targets of Fetal Growth Restriction

The only therapeutic strategy available for the treatment of FGR is the iatrogenic induction of delivery after the administration of corticosteroids for the maturation of the newborn’s lungs and the limitation of perinatal complications. Researchers also try to improve placentation and maternal blood flow to boost fetal growth.

Preclinical and clinical research examines the use of phosphodiesterase type-5 inhibitors (such as sildenafil) to enhance the presence of nitric acid inducing vasodilation. On the other hand, a recent recommendation suggests not to use phosphodiesterase type-5 inhibitors due to an increase in pulmonary blood pressure [193,194], while other trials showed no positive effect on fetal development [162]. Other treatments focus on the uteroplacental circulation, with gene therapy utilizing vascular endothelial growth factor (VEGF) being the principal one [194,195]. VEGF promotes angiogenesis and improves local vasodilation in the uteroplacental circulation, but due to bioethical considerations, it is still experimental. In addition, research is being carried out around the administration of drugs with nanoparticles and microRNAs locally in the maternal vascular endothelium or in the trophoblast. The main goal of maternal vascular endothelial factor gene therapy is to improve local vasodilation and angiogenesis with precise targeting. There are ongoing studies on the efficacy and safety of this targeted approach, as the long-term effects and safety profile are still unknown. Lastly, there are ongoing studies on supplements with melatonin, creatinine, and N-acetylcysteine during pregnancy as possible neuroprotective and cardioprotective factors of the fetus during FGR [196,200,201,202,203,204,205,206]. This technique is called therapeutic angiogenesis [197]. These supplements have the advantage of being non-invasive and relatively safe, but their efficacy is still under consideration, and more clinical trials are needed to confirm the benefits. These investigations are subject to bioethical scrutiny [191,207,208,209,210,211,212]. New research that is still in experimental stage about using nanoparticles in uteroplacental circulation or trophoblast aims to improve uterine and placental blood flow. Melatonin crosses the placental barrier and may be able to protect the developing fetal brain and heart from oxidative stress damage, as it has a strong antioxidant effect [198].

Creatinine, which also crosses the placental barrier, may increase fetal intracellular creatinine and prolong cellular energy homeostasis during hypoxia, possibly thereby protecting the brain and other fetal organs in pregnancies with FGR. N-acetylcysteine can counteract oxidative stress and increase the bioavailability of nitric acid [197]. Each approach needs a thorough evaluation to balance the potential benefits against risks and ethical considerations.

## 14. Conclusions

Globally, FGR affects one out of every ten pregnancies and is a significant issue for obstetric monitoring and care [199,211,212,213,214,215,216]. In total, 10% of fetuses do not grow to the full potential indicated by genetics and population, and the estimated fetal weight is below the 10th percentile. The gestational age at which FGR presents plays a decisive role in the mortality and morbidity of newborns perinatally but also in adulthood. FGR must be identified early so that appropriate prenatal monitoring can be initiated to reduce the likelihood of fetal hypoxia or preterm delivery. This monitoring consists of frequent ultrasound and other examinations, as mentioned above, depending on the severity of the condition. Approximately one-third of pregnancies experience FGR as a result of prolonged placental dysfunction and insufficiency oxygenation of the fetal blood. These changes become permanent and irreversible, with the results being visible before, but especially after, birth. As a consequence, newborns present chronic problems and ailments both in the neonatal period and in childhood and adult life. The choice of time and mode of birth of fetuses with fetal growth restriction is the only treatment available which is still being researched and examined to improve its outcome. Obstetricians have limited treatments for FGR due to the limited trials on pregnant populations, and the limited number of preventive measures for the occurrence of FGR makes it imperative to search for and research new therapeutic approaches. 

## Figures and Tables

**Table 1 jpm-14-00698-t001:** Differences between small fetuses based on various causes.

	Normal Small Fetus	Small Fetus Due to Placental Insufficiency	Small Fetus Due to Congenital Abnormality/Aneuploidy
Fetal anatomy	Normal	Normal	Abnormal
Amniotic fluid	Normal	Reduced	Reduced or increased
Uterine artery pulsatility index	Normal	Increased	Normal
Umbilical artery pulsatility index	Normal	Increased	Possibly abnormal
Middle Cerebral Artery pulsatility index	Normal	Reduced	Normal
Ductus venosus a-wave	Normal	Abnormal	Possibly abnormal

## Data Availability

No new data were created or analyzed in this study.

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
