# Peer review of "Diagnosis, Prevention, and Management of Fetal Growth Restriction (FGR)"

_jpm, 2024, doi:10.3390/jpm14070698_

Round 1

Reviewer 1 Report (Previous Reviewer 1)

Comments and Suggestions for Authors

As attached

Author Response

Dear  Reviewer

  1. We updated the numbering throughout the text to enhance the readability of our article.
  2. We revised the sentence “Three are the categories” as you suggested.
  3. We condensed the conclusion into a single paragraph to better emphasize the key takeaways.
  4. As per your suggestion, we organized our manuscript into fully numbered sections and subsections.

Thank you in advance

Panagiotis Tsikouras

Professor OB /GYN

Reviewer 2 Report (New Reviewer)

Comments and Suggestions for Authors

Your review is thorough and well-researched, providing significant information on the diagnosis, prevention, and management of FGR. The coverage of various factors, diagnostic methods, and management strategies provides valuable insights. However, there are a few areas that could be enhanced to improve the depth and applicability of your study.

1. Emphasize Potential Therapies: i suggest to place greater emphasis on potential therapies for FGR beyond delivery. While acknowledging that delivery is the current primary treatment, it would be beneficial to explore emerging therapies in more detail, highlighting their mechanisms, current research status, and potential for future clinical application.

2. Comparison of Treatment Options: once list the possible treatments, a detailed comparison of different treatment options, even if not fully validated, would add significant value. This comparison should include an analysis of why some therapies have not yet been validated, discussing challenges such as small sample sizes, lack of long-term follow-up studies, and difficulties in conducting randomized controlled trials in pregnant populations.

3.Validation Challenges: Provide a thorough discussion on the reasons behind the lack of validation for certain therapies. 

4. Practical Recommendations:Incorporate practical recommendations for clinicians on managing FGR cases based on the latest evidence. Decision-making algorithms or guidelines for monitoring and intervention would enhance the review’s applicability and provide actionable insights for healthcare providers.

if you believe that talking about therapies is out of your review's scope, I suggest to highlight being very concise on the proper section. just list the fully validated indication to delivery and maybe how they change all over guidelines from different societies. 

Author Response

Dear Reviewer

  1. We believe that emphasizing new therapies is out of our manuscript’s scope.
  2. We expanded upon and added details to the various therapies to enhance their clarity, highlighting the necessity for further research and evaluation to balance potential risks and ethical considerations.
  3. We believe that a comprehensive discussion on the reasons behind the lack of validation for certain therapies falls outside the scope of our manuscript. Instead, we included specific examples within certain therapies to enhance understanding.
  4. In subsection 11.3, we included an algorithm outlining management strategies based on gestational age and Doppler findings to provide practical recommendations, as you suggested. We also added a reference (189).

Thank you in advance

Panagiotis Tsikouras

Professor OB/GYN

This manuscript is a resubmission of an earlier submission. The following is a list of the peer review reports and author responses from that submission.

Round 1

Reviewer 1 Report

Comments and Suggestions for Authors

Comments

The authors reviewed the “Diagnosis, Prevention and Management of Fetal Growth Re-striction (FGR). To make the work better, I have made the following recommendations.

Introduction

The first sentence in introduction is a sweeping statement that has to be backed up by a reference.

Reference this “The evaluation of fetal growth relies on comparing the estimated fetal weight (EFW) with norms that are appropriate for the gestational age.”

The definition of FGR should come earlier in the introduction to guide the reader accordingly

Pathophysiology

In a percentage of women, these changes are not achieved increasing the incidence of a complication during pregnancy. Rewrite as “In a percentage of women, these changes are not achieved, increasing the incidence of a complication during pregnancy

Write in full “PlGF” in the first time of mention.

In Table 1, there was no need abbreviating or using acronyms on the table. Remove the acronyms

Categories of FGR

Rephrase the first sentence as “There are three categories of FGR”

Causes of FGR

This section contains the requisite information, however, stratifying the section into subsections containing the different causes will appeal more to the reader.

Prenatal diagnosis of FGR

Considerations for fetal size, including height, weight, maternal age and ethnicity, fetal sex, and number of pregnancies, are incorporated into normal weight distribution curves that are calculated for particular populations.” Should be referenced

Monitoring FGR

Prevention of FGR

Preventing FGR requires interventions both prior to and throughout pregnancy” can be rewritten as “The prevention of FGR requires interventions both before and during pregnancy.”

Management

Insert a reference here “In cases where a congenital abnormality of the fetus due to a genetic or infectious factor is found, the continuation or termination of the pregnancy is decided based on the gestational age and legislative restrictions of each country, the severity of the abnormality, the expected prognosis and the wishes of the parents, such as the latter arises after full and clear information to the parents by a specialist doctor.”

The management approach discussed here should be discussed in subsections with appropriate numbering to guide the reader

Conclusion

Reference “ Globally, FGR affects one out of every ten pregnancies and is a significant issue for obstetric monitoring and care.”

A paragraph should be enough for the conclusion. Highlight the major takeaways without mentioning unnecessary details here .

General comment

The referencing needs to be improved

For a better flow by the reader, I would better to structure your work in sections and subsections indicated by appropriate numbering.

Do not use acronyms in titles or subtitles. E.g. CPR should not be used. Use the full meaning in the subtitle 

Comments on the Quality of English Language

The English is understandable

Author Response

Comments

The authors reviewed the “Diagnosis, Prevention and Management of Fetal Growth Re-striction (FGR). To make the work better, I have made the following recommendations.

Introduction

The first sentence in introduction is a sweeping statement that has to be backed up by a reference. – added reference

Reference this “The evaluation of fetal growth relies on comparing the estimated fetal weight (EFW) with norms that are appropriate for the gestational age.” - added reference

The definition of FGR should come earlier in the introduction to guide the reader accordingly - corrected

Pathophysiology

In a percentage of women, these changes are not achieved increasing the incidence of a complication during pregnancy. Rewrite as “In a percentage of women, these changes are not achieved, increasing the incidence of a complication during pregnancy – change made

Write in full “PlGF” in the first time of mention. - corrected

In Table 1, there was no need abbreviating or using acronyms on the table. Remove the acronyms - corrected

Categories of FGR

Rephrase the first sentence as “There are three categories of FGR” – rephrase made

Causes of FGR

This section contains the requisite information, however, stratifying the section into subsections containing the different causes will appeal more to the reader. – made subsections for this section

Prenatal diagnosis of FGR

Considerations for fetal size, including height, weight, maternal age and ethnicity, fetal sex, and number of pregnancies, are incorporated into normal weight distribution curves that are calculated for particular populations.” Should be referenced - added reference

Monitoring FGR

Prevention of FGR

Preventing FGR requires interventions both prior to and throughout pregnancy” can be rewritten as “The prevention of FGR requires interventions both before and during pregnancy.” – corrected

Management

Insert a reference here “In cases where a congenital abnormality of the fetus due to a genetic or infectious factor is found, the continuation or termination of the pregnancy is decided based on the gestational age and legislative restrictions of each country, the severity of the abnormality, the expected prognosis and the wishes of the parents, such as the latter arises after full and clear information to the parents by a specialist doctor.” - added reference

The management approach discussed here should be discussed in subsections with appropriate numbering to guide the reader – numbering of the management approach added into subsections

Conclusion

Reference “Globally, FGR affects one out of every ten pregnancies and is a significant issue for obstetric monitoring and care.” – reference added

A paragraph should be enough for the conclusion. Highlight the major takeaways without mentioning unnecessary details here – conclusion edited with highlights

General comment

The referencing needs to be improved – added multiple citations

For a better flow by the reader, I would better to structure your work in sections and subsections indicated by appropriate numbering. – sections and subsections made

Do not use acronyms in titles or subtitles. E.g. CPR should not be used. Use the full meaning in the subtitle – acronyms removed in titles or subtitles

Reviewer 2 Report

Comments and Suggestions for Authors

The authors described a review of a guide to the methods of prevention, diagnosis, and management of pregnancies with FGR, which represents an important cause of perinatal morbidity and mortality. Generally, the presentation is difficult to understand. The flow of the presentation is not organized.

Page 2, in pathophysiology: “With the implantation of the zygote, the creation of new vasculature and the proliferation of trophoblast cells begin, so that before the 10th gestational week, a significant part of the spiral arteries, branches of the uterine arteries adjust to the condition which must be in pregnancy.”

This sentence is not clear. It needs to be rephrased.

Page 2, in pathophysiology: The final part of these vessels does not expand as we would expect; the smooth muscle cells of their wall are not replaced by syncytiotrophoblast cells.

Please provide reference that the smooth muscle cells of the maternal vessels are replaced by syncytiotrophoblast. Is it replaced or influenced by hormone/ others?

Page 2, in pathophysiology again: “..while also regarding penetration of these vessels into the myometrium, does not manage to reach the desired depth and often does not go beyond the level of the uterus.”

What do you mean by does not go beyond the level of the uterus?

The pathophysiology section is not well organized. I feel lost reading through it, perhaps provide subheading in the pathophysiology.

A large portion of pathophysiology is link to apoptosis. How activation of apoptotic pathways relate to FGR should be clearly discussed and established?

Currently, syncytial knot formation in hypoxic condition such as preeclampsia is still poorly understood? Apoptotic or proliferation?

Page 3: Importantly, the number of fibrotic concentrations coexisting with MAC is remarkably higher in FGR pregnancies (45).

What is number of fibrotic concentrations?

Table 1: The term physiological and pathological do not sound correct. Please change to a more suitable term for both.

Page 4, Categories of FGR: In mixed FGR, a decrease in the number as well as in the size of cells is observed. What does this sentence mean? Atrophic and hypoplastic cells? Please elaborate.

Page 5, Causes of FGR: The first paragraph should be in which category, placenta, fetal or maternal?

In Placental causes of FGR, how the oxygen and nutrient are affected? If PE is attributed, some or most PE do not have FGR.

How about VUE, how inflammation results in FGR?

Page 6, Genetic factors: The microRNA section should be brought forward to a new paragraph. It is too general to mention that microRNA is increases in the serum of mother with fetal development problem. Which microRNA? What is the link between the microRNA and fetal growth? What fetal growth disorders? It brings more doubt than understand.

Page 8:  The most common diagnostic method for the differential diagnosis of SGA and FGR is the Doppler of the umbilical artery which evaluates the placental function.

Is the term differential diagnosis correctly used in this context?

Overall, combining the diagnosis, monitoring, prevention and management in FGR has too much information and I feel it is less focus. The review should have focus on pathophysiology and diagnosis alone. While prevention and management can be a separate review. By doing so, the authors can expand further and be more organized.

Comments on the Quality of English Language

English editing service is recommended. 

Author Response

Comments and Suggestions for Authors

The authors described a review of a guide to the methods of prevention, diagnosis, and management of pregnancies with FGR, which represents an important cause of perinatal morbidity and mortality. Generally, the presentation is difficult to understand. The flow of the presentation is not organized. – subsections made

Page 2, in pathophysiology: “With the implantation of the zygote, the creation of new vasculature and the proliferation of trophoblast cells begin, so that before the 10th gestational week, a significant part of the spiral arteries, branches of the uterine arteries adjust to the condition which must be in pregnancy.” This sentence is not clear. It needs to be rephrased. – better rephrase of the sentence

Page 2, in pathophysiology: The final part of these vessels does not expand as we would expect; the smooth muscle cells of their wall are not replaced by syncytiotrophoblast cells. Please provide reference that the smooth muscle cells of the maternal vessels are replaced by syncytiotrophoblast. Is it replaced or influenced by hormone/ others? – better explanation and reference added

Page 2, in pathophysiology again: “while also regarding penetration of these vessels into the myometrium, does not manage to reach the desired depth and often does not go beyond the level of the uterus.” What do you mean by does not go beyond the level of the uterus? – sentence deleted in pathophysiology section

The pathophysiology section is not well organized. I feel lost reading through it, perhaps provide subheading in the pathophysiology. – pathophysiology is divided into subparagraphs

A large portion of pathophysiology is link to apoptosis. How activation of apoptotic pathways relate to FGR should be clearly discussed and established? – the apoptosis is a result of hypoxia which is mentioned in the text

Currently, syncytial knot formation in hypoxic condition such as preeclampsia is still poorly understood? Apoptotic or proliferation? – in page 3 it is mentioned that the syncytial knots are produced due to syncytiotrophoblastic apoptosis

Page 3: Importantly, the number of fibrotic concentrations coexisting with MAC is remarkably higher in FGR pregnancies (45). What is number of fibrotic concentrations?

Table 1: The term physiological and pathological do not sound correct. Please change to a more suitable term for both. – terms changed

Page 4, Categories of FGR: In mixed FGR, a decrease in the number as well as in the size of cells is observed. What does this sentence mean? Atrophic and hypoplastic cells? Please elaborate. – elaborated with a new citation

Page 5, Causes of FGR: The first paragraph should be in which category, placenta, fetal or maternal? – causes of FGR divided in subsections

In Placental causes of FGR, how the oxygen and nutrient are affected? If PE is attributed, some or most PE do not have FGR – The changes in the spiral vessels and the uterine arteries impact the oxygen and nutrient supply, as discussed on page 3 : “The changes in the spiral vessels and the uterine arteries reduce the supply of nutrients mainly in the interlobular space and lead to the induction of oxidative stress, due to the increased oxygen radicals or the reduced amount of antioxidant substances, which occurs in cases of placental insufficiency.”

PE and FGR share common pathophysiological mechanisms, such as abnormalities in the spiral arteries, as explained on page 2 in the first paragraph of the Pathophysiology section.:“These findings are shown in pregnancies complicated by preeclampsia, FGR, recurrent miscarriages, and in some cases, premature births.”

How about VUE, how inflammation results in FGR?

Villitis, an inflammation of the placental villi, can lead to fetal growth restriction (FGR) by disrupting the exchange of oxygen and nutrients between the mother and fetus. This inflammation damages the placental tissue, leading to impaired blood flow and reduced nutrient supply to the fetus, thereby hindering normal fetal growth and development.

Page 6, Genetic factors: The microRNA section should be brought forward to a new paragraph. It is too general to mention that microRNA is increases in the serum of mother with fetal development problem. Which microRNA? What is the link between the microRNA and fetal growth? What fetal growth disorders? It brings more doubt than understand.

It is moved to a new paragraph. Hromadnikova et al. (2012) found that specific microRNAs, such as miR-517a, miR-517c, and miR-518b, are significantly altered in maternal blood in cases of placental insufficiency. These microRNAs play crucial roles in placental development and function. The dysregulation of these microRNAs can contribute to impaired placental function, leading to inadequate nutrient and oxygen supply to the fetus and, consequently, to FGR.

Page 8:  The most common diagnostic method for the differential diagnosis of SGA and FGR is the Doppler of the umbilical artery which evaluates the placental function. Is the term differential diagnosis correctly used in this context? – yes, it is used correctly, as we an obstetrician must differentiate these two conditions

Overall, combining the diagnosis, monitoring, prevention and management in FGR has too much information and I feel it is less focus. The review should have focus on pathophysiology and diagnosis alone. While prevention and management can be a separate review. By doing so, the authors can expand further and be more organized – we want to discuss all the parameters in one review

Reviewer 3 Report

Comments and Suggestions for Authors

Author Response

I would change the sentence “deterioration of the endometrium of the fetus” in

“deterioration of the fetal condition”.

The correction is done

INTRODUCTION

- Introduction line 6: It is not correct to cite only CRL and Estimated fetal weight, because

the evaluation of the fetal growth on charts is based on the evaluation also of biparietal

diameter, head circumference, abdominal circumference and femur length. Please specify

when to use CRL (gestational week) and when to use the other parameters. You report

these data later in the Introduction but I suggest to put them before reference 1.

The correction is done

- Please add this reference between ref.22 and ref. 23: Licini C, Avellini C, Picchiassi E, Mensà

E, Fantone S, Ramini D, Tersigni C, Tossetta G, Castellucci C, Tarquini F, Coata G, Giardina I,

Ciavattini A, Scambia G, Di Renzo GC, Di Simone N, Gesuita R, Giannubilo SR, Olivieri F,

Marzioni D. Pre-eclampsia predictive ability of maternal miR-125b: a clinical and

experimental study. Transl Res. 2021 Feb;228:13-27. doi: 10.1016/j.trsl.2020.07.011. Epub

2020 Jul 26. PMID: 32726711.

The correction is done